# Anabolic Steroids in Fattening Food-Producing Animals—A Review

**DOI:** 10.3390/ani12162115

**Published:** 2022-08-18

**Authors:** Kristýna Skoupá, Kamil Šťastný, Zbyšek Sládek

**Affiliations:** 1Department of Animal Morphology, Physiology and Genetics, Faculty of AgrSciences, Mendel University in Brno, Zemedelska 1, 613 00 Brno, Czech Republic; 2Veterinary Research Institute in Brno, Hudcova 296/70, 621 00 Brno, Czech Republic

**Keywords:** anabolic steroids, skeletal muscle, testes, histological structure, pigs

## Abstract

**Simple Summary:**

Anabolic steroids significantly affect animal tissues and cause morphological and histological changes, which are often irreversible. This issue is currently a very hot topic, as the answers to the questions concerning the health of endangered animals and humans vary greatly from country to country. There is a need to further investigate whether the use of anabolic steroids in animal fattening threatens consumer health and to develop new tools for the detection of anabolic steroids in meat. One possibility for detection could be to observe histological changes in the tissues, which form a typical pattern of anabolic abuse. This review gathered information on the anabolic steroids most commonly used in animal fattening, the legislation governing this issue, and the main effects of anabolics on animal tissues.

**Abstract:**

Anabolic steroids are chemically synthetic derivatives of the male sex hormone testosterone. They are used in medicine for their ability to support muscle growth and healing and by athletes for esthetic purposes and to increase sports performance, but another major use is in fattening animals to increase meat production. The more people there are on Earth, the greater the need for meat production and anabolic steroids accelerate the growth of animals and, most importantly, increase the amount of muscle mass. Anabolic steroids also have proven side effects that affect all organs and tissues, such as liver and kidney parenchymal damage, heart muscle degeneration, organ growth, coagulation disorders, and increased risk of muscle and tendon rupture. Anabolic steroids also have a number of harmful effects on the developing brain, such as brain atrophy and changes in gene expression with consequent changes in the neural circuits involved in cognitive functions. Behavioral changes such as aggression, irritability, anxiety and depression are related to changes in the brain. In terms of long-term toxicity, the greatest impact is on the reproductive system, i.e., testicular shrinkage and infertility. Therefore, their abuse can be considered a public health problem. In many countries around the world, such as the United States, Canada, China, Argentina, Australia, and other large meat producers, the use of steroids is permitted but in all countries of the European Union there is a strict ban on the use of anabolic steroids in fattening animals. Meat from a lot of countries must be carefully inspected and monitored for steroids before export to Europe. Gas or liquid chromatography methods in combination with mass spectrometry detectors and immunochemical methods are most often used for the analysis of these substances. These methods have been considered the most modern for decades, but can be completely ineffective if they face new synthetic steroid derivatives and want to meet meat safety requirements. The problem of last years is the application of “cocktails” of anabolic substances with very low concentrations, which are difficult to detect and are difficult to quantify using conventional detection methods. This is the reason why scientists are trying to find new methods of detection, mainly based on changes in the structure of tissues and cells and their metabolism. This review gathered this knowledge into a coherent form and its findings could help in finding such a combination of changes in tissues that would form a typical picture for evidence of anabolic misuse.

## 1. Introduction

Steroid hormones are generally complex lipophilic molecules synthesized from the precursor cholesterol molecule in endocrine cells in the adrenal cortex, testes, ovaries, and placenta. They are mostly formed as steroid hormone precursors and only after stimulation of the secreting cells of the above-mentioned organs are they converted into active hormones and penetrate from the parent cell by simple diffusion as their intracellular concentration increases. The hypothalamo–pituitary axis is then the superior system for their production and control of endocrine glands by means of tropical hormones. Steroid hormones act on the paracrine and endocrine pathways, where they are released by endocrine cells into the interstitial space and specifically affect cells in the immediate vicinity. In the body, they play a crucial role in regulating and communicating across cells, tissues, and organs throughout an individual’s life [1,2].

The mechanisms of action of individual steroid hormones differ significantly from each other, especially depending on their physical and chemical properties. Hydrophilic hormones act primarily on the cell surface, where they bind to receptors in the plasma membrane. On the contrary, hydrophobic steroid hormones bind to free plasma proteins and are able to diffuse freely across cell membranes and thus activate specific intracellular hormone receptors. The effects of hormones can then be generally divided into long-term and short-term effects. Long-term effects are mediated by high-affinity intracellular receptor proteins that are localized in specific target tissues for each steroid hormone. The interaction of the steroid–receptor complex with hormone-responsive genes leads to tissue-specific expression of proteins that either directly or indirectly generate the body’s biological responses attributed to steroid hormones. This mode of cellular action is generally referred to as the genomic effect of hormones, is slow, and occurs with a time delay of hours or even days [3]. On the other hand, non-genomic action is any mode of action in which gene transcription is not performed directly. These are mostly short-term and rapid reactions that involve specific G protein-coupled receptors located on the cell membrane. They contain many second messengers, including cAMP and diacylglycerol, kinases, and ion flux. Short-term side effects include sexual and reproductive disturbances, fluid retention, increased hunger, general loss of energy, and development or worsening of infections, and severe acne is often reported in humans. Short-term side effects are reversible in most cases [2,4].

Synthetic steroid hormones or anabolic androgenic steroids (AAS) are chemically synthetic derivatives of the male sex hormone testosterone or its structural modifications. Testosterone occurs naturally in the body (especially in males, but to a lesser extent in females) and is involved in the regulation of a large number of physiological processes such as metabolism, development, and regulation of reproductive processes, increasing sexual libido and bone and muscle metabolism. They have a significant anabolic effect on protein synthesis, including muscle tissue, and are therefore applied to animals and humans to rapidly increase muscle mass [5,6]. Steroid hormones can indirectly increase muscle growth through effects on cortisol or by increasing circulating levels of IGF-I to promote long bone growth or directly by affecting the muscle satellite cell population. Some observations suggest an indirect effect through changes in the balance of endogenous hormones. Trenbolone has been reported to increase plasma levels of growth hormone and/or insulin; these hormones are known to stimulate the transport of amino acids across the cell membrane [7,8]. This has the net effect of greater protein synthesis and reduced protein degradation, resulting in increased anabolism of muscle tissue due to larger mature body size and a more active population of satellite cells. Increased proliferative activity of satellite cells should increase muscle growth rate [9]. In addition to anabolic function, they also have androgenic effects because they affect the growth and function of the male reproductive organs. For natural testosterone, both effects are more or less in balance; the individual synthetic preparations differ in their balance between anabolic and androgenic activity. Alkylation of C17, esterification of the hydroxyl group on this carbon, and other chemical modifications seek to increase the anabolic effect and stabilize the molecule, but none of the available anabolic steroids are purely anabolic [5,6,10].

Anabolic steroids have often been used in livestock fattening around the world for their ability to accelerate muscle growth and thereby speed up and ensure fattening and profit from meat sold. Currently, some countries outside the EU approve the use of a number of preparations based on steroid hormones for fattening cattle and sheep. No type of anabolic is approved to promote growth in dairy cows, pigs, or poultry. It is estimated that 80%–90% of cattle in fattening outside the EU are treated with at least one type of growth-promoting anabolic agent. After their possible negative impact on human health began to be investigated, some states began to limit their use and defined it in their legislation [6]. There are still very few studies investigating the anabolic effect directly in farm animals; most studies are conducted on laboratory animals such as mice and rats, or the effects of steroids are investigated in human medicine [5,6]. It turns out that the use of AAS also brings economic and environmental benefits too. Webb et al. determined the environmental and economic impacts of cattle raised with different levels of growth-promoting technologies, non-hormonally treated and implanted. The implanted technology reduced the carbon footprint by 8%, energy consumption by 6%, water consumption by 4%, and reactive nitrogen losses by 8% [11]. A study by Capper et al. evaluated the hypothesis that the use of steroid implants in Brazilian beef cattle would reduce resource use, greenhouse gas emissions, and economic costs of production, thereby improving ecological and economic sustainability. The use of implants reduced GHG emissions for 1.0 × 10^6^ kg HCW of beef by 15.8%. The 6.13% increase in kg of HCW beef produced generates a cost reduction of 3.76% and an increase in the return on invested capital of 4.14% on average [12]. Another study conducted on cattle shows that the use of anabolic implants reduces water consumption by 9.85%, the need for sown land by 9.58%, greenhouse gas emissions by 7.5%, fossil fuel consumption by 4.98%, and animal feed costs by 7.55% [13].

The purpose of the review was to summarize information on legislative measures for the use of anabolic steroids, the most frequently used or abused anabolic steroids, and the effect of anabolic steroids on the tissues of farm animals.

## 2. Legislative Measures for the Use of AAS

### 2.1. History of the Use of Anabolic Steroids

The effects of steroid hormones have been used unknowingly by farmers since ancient times. They noticed increased domestication of animals after neutering and also better meat quality in neutered animals. In ancient Egypt, as early as 800 AD, extract of testicles was used as an aphrodisiac [14,15]. In 1786, the first targeted transplantation of a cock’s testicles was performed, during which it was found that the animal’s secondary sexual characteristics were affected [16,17]. Subsequent experiments in 1849 described changes in appearance and behaviour of cock chickens after castration, and a substance produced by the testes, which is transported in the body by the blood to the target tissues it affects, was described by Arnold Berthold [18,19]. This substance was later identified as the sex hormone testosterone. The anabolic effect of synthetic hormones was first described by Charles Kochakian in 1935 and mentioned the possible use of these substances in tissue regeneration and growth stimulation [15]. From 1937, clinical trials using testosterone derivatives were conducted, and from the 1940s, these substances began to be used by humans as doping agents in sports. In 1948, Dinusson, Andrews, and Beeson conducted the first experiment to support cattle growth with a synthetic estrogen derivative [20]. In 1953, a synthetic ester of estrogen-diethylstilbestrol was first added to cattle feed [21]. The development of anabolic steroid preparations later became a major topic of research and the pharmaceutical industry. AAS have been investigated for possible use in the treatment of a wide range of diseases, such as muscle and wound healing, support in the treatment of osteoporosis, and severe burns, where they speed up tissue regeneration by up to several days. They are prescribed after surgery and radiation therapy and for the treatment of anemia, but their use is currently declining and they are used only after other treatment options have been exhausted. The use of anabolic steroids is also associated with a wide range of negative side effects described above, and, therefore, with any therapy, it is necessary to consider whether their beneficial effect on health sufficiently outweighs the often severe negative effects [22]. In animal production, they took their place in the role of growth stimulants in fattening. Unfortunately, they have been and still are misused in both food-producing animals and by athletes. It also contributes to the fact that there is little information and published texts that 100% confirm the side effects for the final consumers of meat containing residues of anabolic substances that cannot be neutralized. However, many studies based on epidemiological studies draw attention to the relationship of hormonal residues in food with cancer [14,15,23,24,25].

### 2.2. Current Situation in Europe

The use of anabolic steroids to promote growth in food-producing animals has been banned since 1961 in the Netherlands, since 1962 in Belgium, and since 1973 in all Benelux countries. In all countries of the European Union there has been a strict ban on the use of hormonal or thyrostatic substances and growth-promoting beta-agonists since 1988. The only exceptions are some approved veterinary medicines that can be used in defined indications. It is therefore prohibited to use anabolic substances for non-therapeutic purposes in livestock farming, as well as to import meat from animals treated with these substances into the EU market, as an increasing number of studies have demonstrated the possible adverse effects of their residues in meat products on human health. In the past, these rules have limited meat imports from the United States, Canada, Argentina, Australia, Asia, and South Africa, where anabolic steroids are permitted as part of livestock feed and medicines. At present, the import of meat from these countries into the EU is allowed, under the condition of certification that the animals have not been given anabolic substances. The relevant international organizations (FAO, WHO, OIE, Codex) therefore had to set up programs to monitor these substances. The ban on the use of hormones is laid out in Council Directive 96/22/EC [26]. Measures requiring EU countries to monitor these substances and their residues in animals and products of animal origin have so far been laid out in Directive 96/23/EC, which will be replaced by Regulation (EU) 2017/625 of the European Parliament and of the Council from December 2022 [27,28]. Directive (EU) 2017/625 applies to food control and introduces new national surveillance programs for the monitoring of residues of prohibited substances. For example, the acceptable levels of α-testosterone and β-testosterone are set by the FDA at 1.54–2.62 and 1.06–1.56 μg/kg, and the permissible limit for trenbolone is 10 μg/kg in liver and 2 μg/kg in muscle [29]. Meat, milk, eggs, and honey are monitored, and blood, urine, and milk samples are taken from live animals on farms and from carcasses at slaughterhouses. The use of veterinary medicinal products is governed by Council Regulation 2377/90/EC [30,31], which details a procedure for the establishment of maximum residue limits (MRLs) of veterinary medicinal products in foodstuffs of animal origin. The European Medicines Agency decides whether a specified maximum residue limit should be applied to a certain food and lays out rules regarding the conditions for calculating this residue limit. In 2021, a new Directive (EU) 2021/808 was issued, which sets out conservative analytical methods for detecting a few AAS with the well-known organic structure of their molecules [32]. However, it is limited by current knowledge and technical capabilities and therefore cannot cover the detection of newly synthesized AAS and the black market; in large part, producing these new compounds is still a few steps ahead.

In 1999, 2000, and 2002, the Scientific Committee on Veterinary Measures Relating to Public Health (SCVPH) conducted extensive studies on the possible adverse effects of steroid hormone residues in beef and meat products on human health, and some hormones have been declared carcinogens. In the light of these conclusions, Directive 2003/74/EC was adopted with a view to a permanently ban the use of hormones in livestock farming. In 2007, the European Food Safety Authority (EFSA) provided evidence that all hormones banned in the EU may have endocrine, developmental, immunological, neurobiological, genotoxic, and carcinogenic effects, especially in risk groups such as children [33].

Although screening anabolic steroids in pigs and cattle reveals very few positive cases in EU countries, their misuse cannot be ruled out. Steroid hormones are legally produced for therapeutic purposes and can then be diverted to illegal applications [34].

### 2.3. Current Situation in the Rest of the World

In contrast to strict bans in the EU, in countries outside Europe, anabolic steroids can be used legally as fattening aids because national authorities have recognized their use as risk-free or, in some cases, have only defined their use.

In the United States, the Food and Drug Administration (FDA) has approved a number of steroid hormonal preparations for fattening cattle and sheep since the 1950s [35]. Permitted substances include natural testosterone, estrogen, progesterone, and some of their synthetic derivatives. All information on FDA-approved products that can be used as hormonal implants can be found in the Code of Federal Regulations (CFR). All these substances are applied in order to increase the growth rate of animals and provide higher efficiency of feed conversion into meat. New substances are always approved by the FDA only after studies have shown that food from such treated animals is safe for consumers and does not harm the treated animal or the environment. Based on scientific data, the FDA sets limits on the content of hormones in meat. A safe level for human consumption is one which, based on extensive scientific studies, is not expected to have any harmful effect on humans. Hormones are most commonly applied as skin implants from the back of an animal’s ear, and unless otherwise specified, only one implant is administered to the animal at each stage of growth. The ears are removed at slaughter and are therefore not used for consumption. The implants are intended for cattle and sheep only and are freely available in the United States. It is estimated that up to 90% of cattle for fattening have at least one anabolic implant, mostly combined implants. No implant type is approved for accelerating the growth of dairy cows, pigs, or poultry [36]. More than 20 states in the United States that use hormone implants report that their use has reduced greenhouse gas emissions and water and energy consumption in beef production compared with beef grown without growth-promoting hormones [37,38].

Similar laws apply in Canada, where limited amounts of steroid hormones have only been used in fattening cattle since the 1960s [39].

Argentina allows the use of testosterone as well as the synthetic substances zeranol and trenbolone acetate in the fattening of cattle and sheep. In Brazil, since 1991, the use of natural and artificial substances for the purpose of animal growth as well as their import, production, and marketing have been prohibited [40].

In China, the first version of the regulation governing the administration of veterinary medicines, including hormonal products, was issued in 1987 [41,42]. Major amendments were made in 2001 and the current version of the Regulation on the administration of veterinary medicines in China was adopted in 2004. Over time, the Ministry of Agriculture of China (MOA) added supporting regulations as measures for the registration of veterinary medicinal products [40,41,42,43,44], Including administrative measures on over-the-counter veterinary medicinal products and standards for the use of veterinary medicinal products in beef and dairy cattle, pigs, sheep, hens, and rabbits [45]. Since 1999, the MOA has introduced mandatory monitoring and control of the use of veterinary products, including some AAS, each year, and in 2002 issued the latest standards for MRLs for veterinary medicines used in food-producing animals. Most current MRLs are inspired by and derived from EU and US standards, but these standards are often more stringent than those in China. Compared with the USA and European countries, the residue limits of veterinary drugs in China are updated very slowly. Appropriate standards should be speeded up, especially for those medicines that have not yet been included in current standards, in particular antibiotics and large amounts of anabolic hormones [42,46].

In Japan, regulations are stricter and residue limits for veterinary drugs are set similarly to the United States. Anabolic hormone residues must be below the MRL; otherwise, they are considered a potential risk to human health [42,47]. However, studies often indicate that little or no attention is paid to monitoring the misuse of anabolic steroids in fattening farms on small businesses and farms, and therefore steroid hormone abuse may still be encountered in some Asian countries, e.g., Bangladesh and Oman [48,49].

In Australia, the use of steroid hormonal products is partially permitted and has been approved since the mid-1970s. Only Tasmania banned their use in 2000 [47]. The use of hormonal products is overseen by the Australian Pesticides and Veterinary Medicine Authority (APVMA), which states that the use of hormonal growth promoters is safe when applied according to the label instructions. Australia also has a hormone-free food-producing animals program that produces meat for the European market [50,51].

In South Africa, some growth promoters, such as testosterone, estradiol, progesterone, zearanol, and trenbolone acetate, have been approved to promote growth in cattle since 1947 [52]. However, most African countries have laws regulating the use of growing promoters in food-producing animals, but do not have competent authorities to carry out risk analyses of residues of these substances, mainly due to a lack of trained staff or a lack of financial resources [52,53].

## 3. The Most Commonly Used or Misused Anabolic Steroids

### 3.1. Boldenon

Boldenone, a 1(2)-dehydrogenated analogue of testosterone, differs chemically from testosterone by a double bond between C1 and C2. It is characterized by high anabolic and low androgenic activity and due to its effects muscles grow slower but is of better quality than most testosterone derivatives [54,55,56]. It is approved as a veterinary drug in many countries, especially for the treatment of horses, calves, and lambs [57,58]. The use of boldenone is banned by humans and food-producing animals and for doping in racehorses. The most commonly used derivative is Boldenone Undecylenate, which is applied intramuscularly by injection. Under more recent IFHA and FEI laws, steroids can only be administered to a horse under certain therapeutic conditions and the horse must not compete for at least 60 days afterwards [54]. Boldenone has been classified by the International Agency for Research on Cancer as a possible carcinogen, with a higher carcinogenicity index than other used anabolics [54,59]. Boldenone has often been detected in a number of biological samples in various EU countries—Italy [54,59,60], Belgium [54,61], and the Netherlands [60]. However, boldenone and its metabolites are naturally produced steroids by farm animals and are excreted in large amounts in faeces. In addition, boldenone has been shown to be formed from plant fat phytosterols in feed [57,62,63]. Nielen et al., 2004 [64] showed that the presence of 17β-Boldenone conjugates at any level in the urine of calves is evidence of illegal application. Urine must be collected without faecal contamination to avoid erroneous results. Around the year 2000, the frequent use of boldenone in cattle fattening and doping in horses was thus proven [54,59,60]. New studies from 2022 prove that when using ultrasensitive detection methods, we can always detect boldenone in low concentrations in horse urine samples, and for anti-doping tests at horse races the exact spectrum of biomarkers must be used [65]. In stallions, only testosterone and 17β-boldenone are generally considered to be endogenous. However, the ‘semi’-endogenous presence of 17β-boldenone and related compounds, for example in mares and geldings, is a complicating factor in doping control. The IFHA thus abandoned the zero-tolerance policy for stallions and a threshold value for free and conjugated boldenone of 15 ng/mL was established. Despite this threshold for stallions, the presence of 17β-boldenone in the urine of mares or geldings is still prohibited [54,66].

### 3.2. Chlortestosteron

Chlortestosterone, clostebol, is a 4-chlorinated derivative of testosterone. Chlorination prevents the steroid from being converted to dihydrotestosterone and at the same time makes it unable to convert to estrogen. Its use is prohibited in humans and animals, with the exception of some approved veterinary drugs that can be used in specific indications, mainly for dermatological use [67,68]. In the past, it was used mainly for fattening cattle and to increase the performance of racehorses. After 1990, a large number of positive results for 4-chlortestosterone acetate were reported in France and Belgium [69], and therefore efforts began in Europe to develop more accurate and sensitive detection methods [70,71]. However, it is still used for fattening cattle in China and Japan [72].

### 3.3. Nandrolon

Nandrolone, 19-nortestosterone, is an anabolic steroid with a chemical structure similar to testosterone. Due to the lack of a methyl group at C19, it has a better binding affinity for androgen receptors and an increased rate of onset of anabolic activity, and thus also faster effects on muscle growth. Nandrolone and trenbolone have the highest anabolic:androgenic ratio of all AAS. It is most often used in the form of esters, especially as nandrolone decanoate and nandrolone phenylpropionate [73]. Illegal use of nandrolone and its esters in livestock fattening as growth promoters has often been reported in the EU in the past [60,74,75], as nandrolone has long been considered only a synthetic AAS. Its natural form was first confirmed in veterinary plasma medicine in horses in 1980 [76] and subsequently endogenous nandrolone was also confirmed in human medicine in 1984 [77]. 17β-19-nortestosterone was later shown to occur naturally in the urine of boars [78], bulls [79] and sheep, cows, sows, and mares during pregnancy [43,59], often at relatively high concentrations, and evidence of nandrolone abuse is not easy to find [79]. Due to its properties, nandrolone is also often part of anabolic cocktails, so the methods for its detection must be constantly improved [80].

### 3.4. Stanozolol

Stanozolol is a synthetic 17α-alkylated derivative of 5α-dihydrotestosterone. It was previously widely used in human and veterinary medicine, especially for its very high ratio of anabolic: androgenic activity, which means that it has anabolic activity with minimal androgenic undesirable effects [81]. Currently, it is only used in domestic animals, especially dogs, to increase muscle mass and stimulate the appetite of exhausted animals or in the treatment of tracheal collapse [82]. It is not used in livestock for human consumption [83,84]. It is considered a controlled substance, but is gradually being withdrawn from circulation in a large number of countries. In the past, its frequent misuse in animal fattening has been found across European countries. Most of these illegal applications were discovered around 1999, the same year as laboratory methods rapidly improved [85,86]. After administration, stanozolol is metabolized very rapidly, so urinary levels of the parent compound are very low. Thus, the level of urinary excretion is also low, i.e., 3%–5% of the total administered amount, and stanozolol can be detected in urine for 2–3 days. Its detection in urine is therefore almost impossible [87]. Improvements in stanozolol residue analyses are now being achieved by screening for its marker metabolite 16β-hydroxystanozolol [86,88].

### 3.5. Trenbolon

Trenbolone is an androgenic anabolic derivative of nandrolone, specifically nandrolone with two added double bonds in the steroid nucleus. It is most often used in the form of esters, especially as trenbolone acetate (TBA) and trenbolone enanthate (TBE). TBA is a very strong anabolic steroid with strong anabolic effects and therefore has great potential for use in fattening. A number of side effects of its use have been described, including aggression, increased blood pressure and cholesterol levels, skin rashes, negative effects on the thyroid gland, decreased sexual function, and testicular atrophy. Most of these effects are irreversible [89]. TBA is licensed as a growth stimulant for cattle in several countries around the world, such as the United States, Australia and New Zealand, which are major meat exporters [36,90,91]. However, this causes global problems, as meat from these countries often demonstrates a positive TBA above the MRL (10 μg/kg in liver and 2 μg/kg in muscle), even though the meat has been certified as hormone-free [92,93]. In 2000, an extensive inspection of beef in supermarkets imported from Australia and the USA was carried out in Indonesia, and up to half of the samples contained trenbolone residues [94]. The study by El Shaid et al. [95] reported similar results from analyses of sausages and burgers in Egypt, where, again, up to half of the samples were positive for trenbolone.

## 4. Effect of Anabolic Steroids on Tissues

Regular intake of anabolic steroids causes changes in the organization at genetic, metabolomic, and structural levels directly in animal tissues. Increased protein synthesis, fat loss, and a better sensory profile of final meat products are the targeted changes for which anabolic steroids are used in animal fattening. Like natural hormones, synthetic steroids imitate hormonal functions and interfere with the endocrine system of animals. In addition to endocrine changes, steroids have been shown to have carcinogenic, immunotoxic, mutagenic, and teratogenic effects, and the changes are often irreversible. The risks always depend on the age, sex, and individual tolerance of the animals to these substances [96,97]. As the use of steroid substances in fattening animals poses a potential risk to consumers, their presence needs to be monitored and new detection methods developed to prevent the prevalence of public health risks. Many anabolic substances that can theoretically be misused in meat production have not been studied for toxic effects on humans yet [98,99].

### 4.1. Effect of Anabolic Steroids on Skeletal Muscle

Skeletal muscle is the most economically important tissue in animal production and the application of anabolic steroids is mostly targeted at it. The increase in body parameters is best processed in beef cattle after the use of steroidal implants. Implanting increased average daily gain by 21% and improved feed efficiency by 11% in feedlot cattle. In addition, carcass weight was increased by 7% due to implanting. Moreover, a 5% increase was also reported in ribeye size, a 7% decrease in fat cover, a 5% decrease in marbling score, and a 17% decrease in percent of carcasses grading [7]. A study by Perry et al. compared the effects of steroid implants on performance and carcass composition in Holstein, Angus, and Angus x Simmental steers. Daily gain was increased by 17%, 26%, and 21% in Holstein, Angus, and crossbred steers, respectively. The implant increased total daily protein and fat gain by 23% [100].

Most studies describing the effect of anabolics on muscle mass have been performed in animal models such as mice or rats, and from livestock in cattle or sheep [101]. Very few studies have been performed on pig models, although in many European countries pork is consumed to a much greater extent than beef and lamb [102].

What the studies agree on is that AASs increase muscle hypertrophy and protein synthesis, metabolize fat stores, and enhance the response when AASs are combined with strength activity. Thus, in general, the application of steroids leads to an increase in the growth rate of lean mass. At the histological level in muscle, the diameter and area of muscle fibers increase and there is a significant increase in the proportion of fast low-oxidation fibers [103,104,105]. Body weight gain is increased through nitrogen retention and net protein gain without any changes in the digestibility of nitrogen intake. The amount of protein production can then be measured by comparing the amount of nitrogen supplied with the amount of nitrogen in the animal waste. Anabolic hormones therefore accelerate nitrogen retention in the body. However, high doses of AAS can cause degenerative changes in muscle that can lead to loss of function and permanent damage, such as muscle fiber breakdown, cell infiltration, vacuolation, swelling, and mitochondrial damage [105]. The hypertrophy of these multinucleated muscle fibers is accompanied by an increase in the number of myonuclei, with the primary source of new myonuclei being activated and incorporated satellite cells otherwise located outside the muscle fibers. Satellite cells also need to maintain their reserve fund to be able to self-renew. The increased number of satellite cells in the growing muscle causes an increased capacity for skeletal muscle growth [106] and also increases the ability to regenerate muscles. It is demonstrated that AAS supplementation after injury in mice increases the number of proliferating satellite cells and the cross-sectional area of the fibers in the regenerating muscle [107,108]. In addition, the results of new studies show that a common feature of the effects of various growth hormones on muscle cell proliferation and differentiation is the activation of the polyamine biosynthesis pathway [106]. In older animals, the AAS-induced muscle growth processes are slower because most of the remaining satellite cells become silent and no longer multiply or differentiate unless stimulated by injury or strength training [109]. However, recent research [110] shows that faster muscle (re) growth in older animals may be induced by previous AAS use. Myonuclei, acquired during anabolic-induced hypertrophy, are at least partially preserved, and later muscle growth after AAS stimulation is then up to twice as fast. This AAS-influenced “muscle memory” hypothesis is based mainly on data obtained from rodent experiments, and due to differences in muscle formation between species, the transfer of these results to livestock is ambiguous and difficult [110,111]. A more precise identification of the molecular mechanism by which AAS improves livestock muscle growth efficiency is necessary to develop more effective strategies to improve meat production, but the exact mechanisms are currently unknown.

Our preliminary results from experiments on pigs agree with the results of the previously mentioned studies. After the application of testosterone and nandrolone decanolate, there were changes in the muscle of the pigs in the form of an increase in the diameter and area of the muscle fibers and a higher amount of connective tissue (endomysium) between the muscle fibers (Figure 1). Whether the number of nuclei is a statistically significant parameter is still under study, as the counting of nuclei in histological sections is not accurate, as they are randomly distributed along the fiber.

### 4.2. Effect of Anabolic Steroids on the Male Reproductive System

The AAS affects the genitals, especially in males, in a completely different way than the muscles. Thus, they do not cause their growth, but numerous studies have shown that AASs have a strong toxic and degenerative impact [108,112,113,114]. The application of anabolics has an impact mainly on the testicles, where it causes disturbances of spermatogenesis, epithelial degeneration, and overall disorganization of the anatomical and histological structure. The membranes of the seminiferous tubules are deformed, the layers of the germinal epithelium inside the tubules are reduced, and the number and mobility of sperm are reduced [113]. This loss of germ cells causes mitosis and meiosis to stop, and high doses of AAS cause degeneration and decreased Sertoli cell support function. As a result, the layers of sperm cells are reduced to one layer and the sperm cells begin to degenerate [112]. Interstitial spaces expand, necrotic Leydig cell numbers increase, and overall anabolic uptake has an inhibitory effect on the hypothalamic–pituitary testicular axis, leading to a reduction in natural testosterone production. There is also damage to the additional gonads and enlargement of the prostate [114,115]. Ultimately, AAS has similar effects to chemical castration. Several studies suggest a positive relationship between AAS application and infertility and carcinogenesis progression. Commonly used hormones, such as nandrolone and stanozolol, can potentially induce the progression of various cancers, such as Leydig cell tumors, through many pathways [109,116]. Approximately six weeks after cessation of anabolic hormone therapy in a mouse study, a gradual improvement in the structural condition of the testis was observed, but there was no complete recovery of function [114,117].

Our preliminary results from experiments on pigs show a reduction in the germinal epithelium in the seminiferous tubules after application of nadrolon to complete destruction of the epithelium and the formation of a fibrous network from the ligament inside the lumen of the canals, reminiscent of prepubertal structures. There was a decrease in the incidence of mature spermatids and spermatozoa. The seminiferous tubules were spaced apart. Leydig cells deformations, wrinkling, and degeneration occurred (Figure 2). Thus, reduced testosterone production can be expected, leading to reduced fertility or even complete sterility of individuals.

### 4.3. Effect of Anabolic Steroids on Other Tissues

In addition to the above-mentioned tissues, in which the manifestation of AAS is the most intensively studied, anabolics also affect other tissues in the body. These are mostly significant degenerative changes and disorganization of the histological structure. A negative effect on the heart muscle, kidneys, liver, and bones is described [114]. The results of numerous studies, especially in rats, agree that long-term use of AAS causes pathological cardiac hypertrophy that persists and even increases after treatment. After higher doses of synthetic steroids, such as nandrolone decanolate, vasocongestion, muscle fiber retraction, massive dilaceration, and muscle fiber rupture occur in the heart tissue. Reduced pumping efficiency of the heart is reported [109,114,118]. These pathophysiological cardiac effects can be linked to the fact that androgen receptors in cardiomyocytes allow steroids to affect the physiology of the heart [114].

Persistent changes even after discontinuation of AAS treatment are also described in the kidneys. After application of testosterone esters and nandrolone decanoate, swelling and weight gain occur in the kidneys. On the contrary, there is atrophy and deformation of the glomeruli and cracking of the glomerular walls. Damage to the urinary tubules, rupture, vacuolar epithelial degeneration of the proximal coiled tubules, and thickening of the basal lamina in the distal coiled tubules are visible. The tubules move away from each other and bleeding occurs between them due to overloading of the renal vessels [114,115,119,120]. After long-term application, lesions and large necrotic areas form in the urinary tubules after a few weeks [114]. These findings suggest the possibility of chronic kidney damage after AAS application, which may lead to progressive kidney failure. The study by Cho et al. [121] indicated that AAS could be a risk factor for the development and progression of renal cell carcinoma. These changes in renal structure also correlate with the finding of frequent renal disorders in bodybuilders who have used high doses of AAS. Hartung et al. 2001 [122] stated that renal biopsy reveals nephrosclerosis and severe kidney lesions in bodybuilders.

In the liver, in contrast to the kidneys, the greatest weight gain was observed at low doses of nandrolone decanolate. At higher doses there was only a slight increase, but there was a sharp increase in the amount of liver enzymes. There were disorders of bile formation and drainage, dilation of blood vessels, and their ruptures [120,123].

AAS and androgens are generally involved in bone growth and maintenance of bone homeostasis [108]. Studies show that anabolics do not affect bone quality, but speed up the strengthening process and increase bone callus formation after injury [124]. Souza et al. [125] described that after administration of nandrolone decanolate, there was no increase in femoral weight or length and there was no difference in relation to the diameter of the epiphysis and the diaphysis. Marchi et al. [126] found that nandrolone stimulated bone marrow production and it could be used in the treatment of aplastic anemia. In sheep, the beneficial effect of stanozol on articular cartilage regeneration in femorotibial osteoarthritis has been confirmed [127]. The positive effect of stanozol on joints is also described by athletes and, the study by Falanga et al. [128] disclosed that stanozol increases collagen synthesis. However, this effect has not been described with other anabolic steroids.

## 5. Conclusions

Anabolic steroids significantly affect animal tissues and cause morphological and histological changes, which are often irreversible. This issue is currently a very hot topic, as the answers to questions about endangered animal and human health vary greatly from one country to another. Whether the use of AAS in animal fattening threatens the health of consumers needs to be further studied and new tools for detecting AAS in meat need to be developed. One of the detection options could be the observation of histological changes in the tissues that form the typical picture of AAS misuse.

## Figures and Tables

**Figure 1 animals-12-02115-f001:**
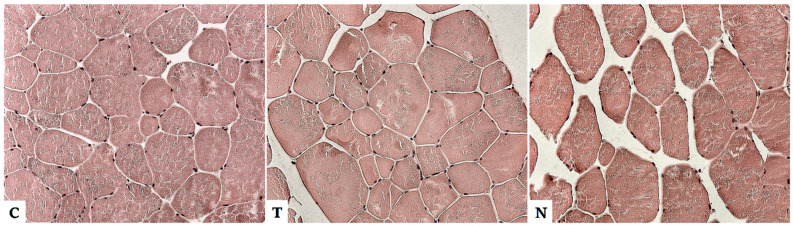
Histological section of muscle fibers of pig from the control group without anabolic steroids (C), from the group after testosterone administration (T), and from the group after administration of nandrolone (N). After the application of AAS, there was an increase in the diameter and area of muscle fibers and a greater amount of endomysium between muscle fibers in the muscle of pigs. With nandrolone, the steroid effect was even stronger (unpublished data).

**Figure 2 animals-12-02115-f002:**
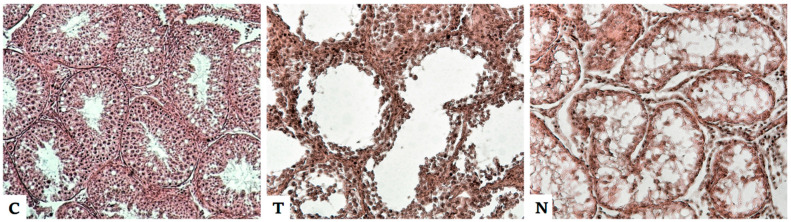
Histological section through the seminiferous tubules of pig testes from the control group without anabolic steroids (C), from the group after testosterone administration (T), and from the group after administration of nandrolone (N). A reduction in the germinal epithelium in the seminiferous tubules after testosterone, the destruction of the epithelium, and the formation of a fibrous network from the ligament inside the lumen of the canals after nandrolone application are evident (unpublished data).

## Data Availability

Not applicable.

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
