# Peer review of "Anabolic Steroids in Fattening Food-Producing Animals—A Review"

_animals, 2022, doi:10.3390/ani12162115_

Round 1

Reviewer 1 Report

Comments to the Authors of manuscript number: animals-1804214 entitled “Anabolic Steroids in Fattening Food-Producing Animals – a Review”.

The review summaries and presents data relating to anabolic steroids. Authors present current information about their use along various countries in the word with many elements of history. Authors also involve some data from the study performed by themselves. It is clever way to present own results, especially when the results are from the funded study. The review is very interesting, and is worth to publication. The paper is very well prepared. However, I have some questions:

1. L 48 -what is mother cell? This term should be changed.

2. It is good to mention that they are produced in the response to the hypothalamus-pituitary axis.

3. L49- they act on the endocrine and paracrine pathway

4. L90 – the reference should be given

5. L 93- “Soma KK. Testosterone and aggression: Berthold, birds and beyond. J Neuroendocrinol. 2006 Jul;18(7):543-51. doi: 10.1111/j.1365-2826.2006.01440.x. PMID: 16774503; PMCID: PMC2954190.”

6. L 154- reference add, please

7. L 177 – reference is needed

8. L200 – reference is needed

9. L 213 - 1(2)-dehydrogenated analogue of testosterone

10. the part of 3.5 – it is worth to mention all side-effects of its use like aggression, the negative impact on thyroid gland and others

11. L 302 – reference should be added

12. L 335 – it is not clear. What studies?

13. L338- what about connective tissue?

14. L 350- disturbances

15. L 352- reference should be added

16. Figure 2- it need the description according to the pathology given in L367-374

17. L 383- reference should be added

18. L 410- the maintenance of bone homeostasis

Reviewer 2 Report

The use of anabolic steroids is a relevant topic within production animals since they are not only associated with side effects in animals, but also in consumers. In recent years, with the growing interest in opting for products of animal origin with high quality and natural, the use of added substances can be a decisive factor. This article provides relevant information on the use and pharmacology of anabolics. One area of ​​opportunity to consider is the cost-benefit effect of anabolic steroid use. The main weakness of this article is the lack of a clear objective, in addition to the fact that the discussion of various countries is mentioned at the beginning of the document but is not discussed throughout the manuscript. Likewise, the description of the different anabolic is very superficial. I have left some comments that might help the authors.

Line 25: Please, mention some examples of damages to tissues or organs and the main alterations present in the reproductive system.

Line 51: I would suggest including a paragraph regarding the current use of steroid hormones in production animals. The authors could briefly explain why they are used (e.g., as growth promotors), in which species, percentages of use according to the species or countries. In this way, the reader can understand the importance of anabolic in food-producing animals.   

 Line 62: Specify what are the short-term effects that steroid hormones can cause.

 Line 82: Apart from the pharmacologic explanation of steroid hormones, I considered relevant to mention the limited information regarding the damage to the skeletal muscle or reproductive system cells. Also, I suggest including at the end of the introduction the objective of the present review, mentioning the main species the authors included in their review.

Lines 102-103: Could the authors include in which diseases AAAs are used as treatment? Additionally, although it is mentioned that there is a “large number of harmful side effects”, I would suggest specifying the side effects on animals or in humans eating those animals.

Line 113: Please, add why anabolic substances were prohibited in veterinary medicine. If it was after the recognition of adverse effects or why it is not permitted to use anabolic for non-therapeutic purposes.

 Line 118: Generally, to export animal-derived products in European countries it is necessary the lack steroid hormones. Therefore, this aspect is not only related to European countries but to those that export products or have commercial relations with these sites.

Line 126: It would be interesting to mention if there is a permitted level of anabolic substances and the approved methods to detect prohibited substances.

 Line 140: I think it is relevant to add the adverse effects that steroids can cause on human health apart from being carcinogens.

 Line 144: Please, insert a reference.

 Lines 157-159: In the introduction section the authors could include the physiological basis of why steroid hormones promote growth and efficiency of feed conversion. A table comparing the doses and growth rate could also help to understand the use of anabolic in production animals.

Line 192: Please, add references or include more studies regarding this issue. Discussing the situation in China cannot be used to generalize the situation in every Asian country.

Line 197: Clearly state the percentage when mentioning “therefore the percentage of misuse is still quite high”.

 Line 210: Please, consider including a section about the cost-benefits of steroid hormone use, mentioning the environmental benefits that it may have. Also, please, discuss the current legislative situation in Argentina. In the abstract, the authors mention this country, but the legal aspects of this country are not addressed. The same applies to Canada.

Lines 213-231: In this section, I consider it relevant to include which food-producing species boldenone is still misused since it is the main topic of the present article. It is mentioned that farm animals can naturally produce boldenone, is there a concentration limit to consider in, for example, racing horses? The authors could include current doses or administration routes to understand the pharmacological management of this drug.

 Line 236: Instead of leaving a general idea about the “specific indications” where chlortestosteron can be used, please, specify in which situations this hormone is still used (e.g., if in some countries is still used for fattening cattle or racehorses)

 Line 256: I suggest including the current detection methods that are used, mentioning the sensibility or limitans of these methods, not only for nandrolone but for other steroid hormones.  

Lines 261-262: Please, add examples of the current usedof stanozolol in small animals (e.g., that is used as a treatment for tracheal collapse or to promote appetite).

Lines 266-270: Please, modify the sentence: “stanozolol is metabolized very rapidly, so that urinary levels of the parent compound are very low” by putting the time that takes to metabolize the hormone, and also mention what is considered low urinary levels.

 Line 279: What is the maximum residue limit for trenbolon? Please, include this.

 Line 282: Please, delete the year for El Shaid citation (check citation style of the journal).

 Line 302: It would be adequate to include exact data regarding feed conversion and anabolic. Some articles that may help the authors are Perry (10.2527/1991.69124696x), who mentions that it can cause an increase of 16% in weight gain, while Peters (https://doi.org/10.1017/S000335610004157X) states a 28%.

Line 304: Please, insert a reference.

 Line 345: Could the authors give more information about the observed changes and the possible biological explanation of the increase in the diameter and of the muscle fibers?

Line 366: Do these changes been documented in production animals? If this is the case, I suggest including in which species this has been reported.

Round 2

Reviewer 2 Report

Agradezco a los autores por haber hecho mis sugerencias. El artículo ha mejorado mucho y ya está listo para ser publicado. Sólo resta a los autores hacer los ajustes que el Editor sugiere.

No tengo comentarios adicionales.

Author Response

Dear reviewer.
We thank you for evaluating our changes and corrections.

Since you didn't request any further changes yourself, we didn't change anything further.

Thank you.